# SCDNet: A Deep Learning-Based Framework for the Multiclassification of Skin Cancer Using Dermoscopy Images

**DOI:** 10.3390/s22155652

**Published:** 2022-07-28

**Authors:** Ahmad Naeem, Tayyaba Anees, Makhmoor Fiza, Rizwan Ali Naqvi, Seung-Won Lee

**Affiliations:** 1Department of Computer Science, University of Management and Technology, Lahore 54000, Pakistan; f2019288007@umt.edu.pk; 2Department of Software Engineering, University of Management and Technology, Lahore 54000, Pakistan; tayyaba.anees@umt.edu.pk; 3Department of Management Sciences and Technology, Begum Nusrat Bhutto Women University, Sukkur 65200, Pakistan; makhmoor.fiza@bnbwu.edu.pk; 4Department of Unmanned Vehicle Engineering, Sejong University, Seoul 05006, Korea; 5Department of Data Science, College of Software Convergence, Sejong University, Seoul 05006, Korea; 6School of Medicine, Sungkyunkwan University, Suwon 16419, Korea

**Keywords:** transfer learning, biomedical image, automated/computer aided diagnosis, melanoma, skin cancer

## Abstract

Skin cancer is a deadly disease, and its early diagnosis enhances the chances of survival. Deep learning algorithms for skin cancer detection have become popular in recent years. A novel framework based on deep learning is proposed in this study for the multiclassification of skin cancer types such as Melanoma, Melanocytic Nevi, Basal Cell Carcinoma and Benign Keratosis. The proposed model is named as SCDNet which combines Vgg16 with convolutional neural networks (CNN) for the classification of different types of skin cancer. Moreover, the accuracy of the proposed method is also compared with the four state-of-the-art pre-trained classifiers in the medical domain named Resnet 50, Inception v3, AlexNet and Vgg19. The performance of the proposed SCDNet classifier, as well as the four state-of-the-art classifiers, is evaluated using the ISIC 2019 dataset. The accuracy rate of the proposed SDCNet is 96.91% for the multiclassification of skin cancer whereas, the accuracy rates for Resnet 50, Alexnet, Vgg19 and Inception-v3 are 95.21%, 93.14%, 94.25% and 92.54%, respectively. The results showed that the proposed SCDNet performed better than the competing classifiers.

## 1. Introduction

Skin cancer is caused by the uncontrolled growth of abnormal skin cells which results in malignant tumors [1]. When these cells are exposed to ultraviolet rays, a mutation occurs in the DNA which affects the normal growth of skin cells and eventually results in skin cancer [2]. Skin cancer accounts for one-third of all cancer cases worldwide, according to the World Health Organization. Skin cancer is a public health issue that affects people all over the world [3]. Dermoscopy is a common technique to detect skin cancer. However, the initial appearance of multiple types of skin cancers is the same so it is very challenging for the dermatologist to identify them accurately [4]. The average accuracy of dermatologists is between 60% and 80% for skin cancer diagnosis using dermoscopic images [5]. The dermatologist with 3 to 5 years of experience has a claimed accuracy of above 60%. The accuracy rate drops for a less experienced dermatologist. However, a dermatologist with ten years of experience has an accuracy of 80% [6]. In addition, dermoscopy performed by untrained dermatologists may impair the accuracy of skin cancer detection [7,8]. Extensive training is the major requirement for dermoscopy [9]. Numerous forms of skin cancers can be identified using dermoscopic images. However, melanocytic and nonmelanocytic are two main types of skin disease. The melanotic consists of melanoma and melanocytic nevi. Whereas, basal cell carcinoma (BCC), squamous cell carcinoma (SCC), vascular (VASC), benign keratosis lesions (BKL), and dermatofibroma (df) are the type of nonmelanocytic skin diseases [10].

Melanoma is the most common and fatal type of skin cancer which is caused by irregular melanin production in the cells of melanocytes [11]. It is the fastest-growing type of cancer. Melanoma is further classified as benign and malignant [12]. Melanin is typically present in the epidermal layer in benign lesions (common nevi) [13]. In the malignant lesion, melanin is produced at an abnormally high rate. Every year more than 5 million new cases of skin cancer are registered in the United States [14]. Melanoma is responsible for almost 75% of all skin cancer deaths. In the United States alone every year 10,000 deaths occur due to melanoma [15]. In 2021, 106,110 cases of melanoma were reported in the United States while 7180 people died due to melanoma [16]. It is anticipated that there will be a 6.5 percentage point rise in the total number of deaths caused by melanoma in the year 2022. Moreover, it is anticipated that there will be 197,700 new cases of melanoma diagnosed in the United States in 2022 [17]. Every year, approx. 100,000 new instances of melanoma are detected in Europe [18]. In Australia melanoma is diagnosed in 15,229 people annually [14,19]. Skin cancer incidence rates have risen in the past decade, and the melanoma rate has increased by 255% in the United States; in the United Kingdom it hass increased by 120% since the 1990s [20,21]. However, melanoma is considered as a highly treatable cancer if it is diagnosed at the initial stage. The survival rates are above 96% in the initial stage. However, the survival rates drop to 5% in the advanced stage. The treatment of melanoma becomes difficult when it has spread throughout the body [10]. Early detection of skin cancer depends upon skin color, hair, and air bubbles. Moreover, insufficient medical resources and the high cost of treatment in developing countries delay the early detection of skin cancer [22]. The ABCD rule was used for the early diagnosis of skin cancer. The ABCD rule segments the areas of skin lesions to achieve better diagnostic accuracy. However, the segmentation is based on color channel optimization and the set level technique is not efficient which lowers the overall accuracy [23]. Moreover, researchers prefer a public dataset to the 7-point checklist, which was developed for non-dermatological medical workers [24]. The development of a computer-aided diagnosis (CAD) system resolves the problems of dermatologists. Moreover, the CAD system uses image processing and machine learning techniques for the analysis of dermoscopic images [25]. The diagnosis improves as more data becomes available for computer-aided systems [26]. Therefore, the primary goal of researchers is to develop an artificial intelligent (AI) based diagnosis system capable of identifying and classifying multiple types of skin cancer at an early stage. Moreover, the doctors will early detect the skin lesions by using the machine and deep learning techniques which also reduce the unnecessary surgeries and biopsies [27].

In this paper, a novel method named skin cancer detection classifier network (SCD Net) is proposed for the multiclassification of skin cancer using dermoscopic images. As noted, the proposed method is based on the Vgg16 and convolution neural network (CNN) approaches, achieves exceptionally high accuracy for the detection and classification of skin cancer. This study utilizes dermoscopic images for the automated classification of skin cancer. When skin cancer is detected in its earliest stages, medical professionals have a better chance of initiating treatment on time and avoiding the disease’s progression. Additionally, the performance of SCDNet is compared with the state of the arts medical classifiers which include AlexNet [28], Vgg19 [29] ResNet-50 [30], and Inception v3 [31]. This study classifies the dermoscopy images into the four major classes of skin cancer which include Melanoma, Melanocytic Nevi, Basal Cell Carcinoma and Benign Keratosis.

The following is a summary of the contribution of this study:The authors propose SCDNet which is based on CNN and Vgg16. SCDNet extracts prominent features from dermoscopic images and classifies them into four major classes of skin cancer, namely, Melanoma, Melanocytic Nevi, Basal Cell Carcinoma and Benign Keratosis. Moreover, the performance of the proposed method in terms of accuracy, f1 score, AUC, specificity and sensitivity is also compared with the four well-known classifiers in the medical domain, namely, Inception v3, Alexnet, Vgg19 and resnet50.The developed SCDNet was trained on dermoscopic images collected from the ISIC 2019 dataset [32] which contains images of Melanoma, Melanocytic Nevi, Basal Cell Carcinoma and Benign Keratosis. The proposed method was trained, tested, and validated using images in a 70:20:10 ratio.The leave one out cross validation (LOOCV) is also used to evaluate the performance of SCDNet.The SCDNet showed an exceptional performance by achieving an accuracy of 96.91%, 92.18% sensitivity, 92.19% precision and 92.18% F1 score.AA novel deep learning framework is designed for the diagnosis of skin cancer using dermoscopic images.

The structure of the paper is summarized as follows. The literature review is discussed in Section 2. Section 3 contains the proposed method. The experimentation results and discussion are provided in Section 4 and Section 5. The conclusions of the research are discussed in Section 6.

## 2. Literature Review

A significant amount of research has been conducted for the diagnosis of skin cancer in order to help healthcare experts with early diagnosis of the disease. However, recent studies focus on the automated detection of various types of skin cancers by using different artificial intelligence techniques. 

Moloud et al. [25] proposed a novel Bayesian deep learning method based on a three-way decision theory for the binary classification of skin cancer, considering the level of uncertainty. Moreover, the proposed method applies different uncertainty quantification UQ methods and deep neural networks in classification phases. Two datasets are used for the experimentation of this model. One dataset was collected from Kaggle which contained 2637 images for training and 660 images for testing whereas, the second dataset was ISIC 2019, containing 7234 images for testing and 1808 images for training. The achieved accuracy for the Kaggle dataset was 88.95% and for the ISIC the accuracy was 90.96%. Abbas et al. [33] introduced a custom build model for the classification of skin cancer based on the seven-layer of deep convolution network. The proposed model was trained from scratch. The experimentation was performed on 724 dermoscopic images containing 374 images of benign nevus (BN) and 350 images of acral melanoma (AN) which were collected from the Yonsei University Health System, South Korea. Transfer learning was also used to compare the performance of the model; Resnet18 and AlexNet were fine-tuned and modified to train on the same dataset. The model achieved an accuracy of more than 90%. However, the accuracy reached 97% by using transfer learning approaches. Ismail et al. [34] presented a model which utilizes a hybrid convolution neural network and bald eagle search optimization for the binary classification of skin lesions as melanoma or normal. The experimentation was performed on a publicly available dataset ISIC 2020. The issue of class imbalance was also resolved by applying random sampling and augmentation. The proposed method achieved a remarkable accuracy for the detection of melanoma of 98.7%, sensitivity of 100%, specificity of 96.4% and f-score of 98.40%. Mijwil et al. [35] applied the three architectures (Resnet, Vgg19 and Inception v3) of the convolutional neural network (CNN) model for the analysis of 24,000 images collected from the ISIC archive. This study used many parameters to analyze the best architecture for the classification of benign and malignant images. Among the selected architectures, Inception v3 provided promising results on the data. Inception v3 achieved an accuracy of 86.9%, sensitivity of 86.1%, specificity of 87.6% and precision of 87.4%. Nawaz et al. [36] proposed a fast region-based convolution neural network (RCNN) based on deep learning. This method also utilizes fuzzy k-means clustering for the diagnosis of benign and malignant melanoma images. The experimentation for this method was performed on three publicly available datasets named ISIC 16, ISIC 17 and PH2 dataset. Several preprocessing techniques for noise removal and image enhancement were applied to achieve outstanding results. The proposed method achieved an accuracy of 95.4% for the ISIC 16 dataset, whereas for the ISIC 17 93.1% accuracy was achieved and the accuracy of 95.6% was achieved for the PH2 dataset. Dorg et al. [37] proposed a novel idea in which features are extracted by using a pre-trained Alex net whereas the classification of skin cancer is performed by using ECOC SVM. The experimentation was performed on four different types of skin cancer comprising a total of 3753 images which were collected from the internet. The proposed method achieved an accuracy of 95.1% for squamous cell carcinoma, 98.9% accuracy for actinic keratosis, 91.8% accuracy for basal cell carcinoma and 90.74% accuracy for melanoma. Afza et al. [38] presented a framework that implements deep learning with two-dimensional superpixels. They introduced the color segmented images which were extracted through the segmented lesion mapped on the dermoscopic images. Resnet-50 is used for the feature extraction through transfer learning whereas the naïve bayes classifier is used for classification. The experimentation was performed on the Ham1000, ISIC 2016 and PH2 datasets. This method achieved an accuracy of 85.50% for the Ham1000 dataset, 91.1% accuracy for the ISIC 2016 dataset and 95.40% accuracy for the PH2 dataset. Hameed et al. [39] presented the algorithm named multi-class multi-level (MCML) which classifies the skin lesion into multiple skin disease classifications including healthy, malignant, benign and eczema disease. Traditional machine learning with improved noise removal techniques and progressive deep learning was used to build the proposed algorithm. A total of 3672 images from different sources were used to evaluate the diagnosis efficiency of the proposed algorithm which achieved an accuracy of 96.47%. Lokash et al. [40] introduced a framework named Transfer Constituent Support Vector Machine (TrCSVM) based on transfer learning (TL) for the classification of melanoma from skin lesions using feature-based domain adaption (FBDA). The presented framework comprises of support vector machine (SVM) and Transfer AdaBoost (TrAdaBoost). The ISIC 2017 dataset was used as a training dataset. It originally contained 2000 dermoscopic images which were extended to 50,000 images using data augmentation. A total of 112 features were extracted from the ISIC 2017 dataset. The testing of the proposed framework was performed on six different datasets, which included PH2, HAM10000, MED-NODE, Dermatology Atlas, Dermnet Atlas and Dermis, with the achieved accuracy of 98.9%, 82.2%, 89.5%, 82.1%, 79.2%, 87.7%, respectively. Mehak et al. [41] proposed a method for the multiclassification of skin cancer to implement data augmentation, deep learning and transfer learning. Fine-tuned deep models and augmented datasets were trained via transfer learning. Several machine learning algorithms were used for the classification of selected features; based on accuracy, the best classifier was selected for the skin cancer classification. The proposed framework achieved an accuracy of 92.7% for the augmented HAM10000 dataset. Attique et al. [42] presented a mask recurrent convolution neural network (MASK R-CNN) for lesion segmentation. In this architecture, a feature pyramid network (FPN) along with denset50 is used for the mask generation whereas, the classification of higher features is performed through the 24-layer convolution neural network. The validation of the segmentation process is performed on PH2, ISIC2016 and ISIC2017, and the classification is performed on HAM10000. The achieved accuracy was 86.50% according to the experimentation process. A novel technique for the extraction and classification of hybrid features is introduced by Ibrahim et al. [43]. Wavelet transform (DWT), gray level co-occurrence matrix (GLCM) and local binary pattern (LBP) are the three algorithms used for feature extraction, and the classification of these features is performed by feedforward neural network (FFNN) and artificial neural network (ANN). The efficiency of the proposed method was evaluated using ISIC 2018 and PH2 datasets. The measured accuracy for the ISIC 2018 dataset was 95.24% whereas accuracy achieved by PH2 dataset was 97.91%.

The researchers of [34,35,38,41] only focused on the binary classification of skin cancer whereas a few researchers have worked on the multiclassification of skin cancer. Moreover, previous investigations focused on pre-trained algorithms for the classification of skin cancer. The high accuracy of deep learning classifiers opens a new door for disease diagnosis. The use of convolution neural networks in the health care system has achieved a better diagnostic accuracy for diseases such as skin cancer detection [44], chest infection detection [45], brain tumor detection [46,47], breast cancer detection [48] and the analysis of genetic sequences [49].

## 3. Proposed Methodology

This section presents the experimental techniques used to analyze the performance of the proposed model, as well as four well-known pre-trained models which include Inception-v3, ResNet-50, Alexnet and Vgg-19. For this purpose, we developed a novel method, named as SCDNet, for the multiclassification of skin cancer trained on the widely used ISIC 2019 dataset. The images of the dataset are first preprocessed to reconcile with the input dimension of the architecture utilized in this study. For this purpose, the size of input images is fixed to 224 × 224 resolution. Moreover, the overfitting of the model is prevented by applying the process of normalization on images of the ISIC 2019 dataset. The images of the dataset are split into training, testing and validation sets. To train the SCDNet, different types of cancerous images are used as training and validation sets. The experimentation was run for 50 epochs. After all of the epochs were finished, the proposed model reached the accuracy in training and validation that had been predicted. Moreover, the comparative analysis of SCDNet was performed with the four state-of-the-art pretrained methods in terms of accuracy, f1 score, precision and sensitivity. The SCDNet constructs the output images by combining the prominent features and classifying the images into four major classes of skin cancer, namely, Melanoma, Melanocytic Nevi, Basal Cell Carcinoma and Benign Keratosis. Figure 1 shows the architecture of the proposed SCDNet.

### 3.1. Dataset Description

The primary focus of this research is dermoscopic images for skin cancer due to its high impact across the world [50,51]. The ISIC 2019 dataset is utilized, which contains a large number of dermoscopy images of skin lesions collected from multiple sources. The dataset contains a total number of 25,331 dermoscopic images of different classes of skin cancer. The dataset contains 4522 images of Melanoma, 12,875 images of Melanocytic Nevi, 3323 images of Basal Cell Carcinoma and 2624 images of Benign Keratosis; the remaining images belong to different categories which are not considered in this study [32]. Figure 2 shows the sample of images from the ISIC 2019 datasets. The proposed SCDNet is trained and tested on the ISIC 2019 dataset. to obtain efficient results. The dataset is split in the ratio of 70%:20%:10% for training, testing and validation sets. Table 1 shows the division of the dataset.

### 3.2. Data Normalization and Preprocessing

The steps for pre-processing were kept to a minimum in order for the proposed method to achieve better generalization [52]. The built-in KerasImageDataGenerator was used to perform the basic pre-processing [53]. The dermoscopy images in the dataset have a resolution of 450 × 600 pixels. To reconcile the images with the input of the model we downscaled the resolution of images to 224 × 224 pixels [54]. Moreover, the data normalization technique was also used to ensure that the proposed approach is properly trained [55]. As a result, we prepared our datasets to input into the SCDNet for training purposes.

### 3.3. Pre-Trained Classifiers

In this section existing state of art pre-trained classifiers such as AlexNet, Inception v3, Resnet50 and Vgg19 are applied to classify the different classes of skin cancer using dermoscopy images. The ImageNet (ILSVR) database was used to train all of the classifiers. There are thousands of different objects in the ILSVR dataset which are utilized for the training and to analyze the classification performance of the model [56]. The architecture of Alexnet is free and open-source and it is used for various research contexts. The architecture uses five convolutional layers, three max-pooling layers, two normalization layers, two fully connected layers, and a softmax layer. The max-pooling operation is carried out using the pooling layers. Every convolutional layer has its own set of convolutional filters in addition to a nonlinear activation function known as ReLU; because there are entirely linked layers, there is no variation in size. The succeeding three-step process also uses a single max-pooling layer of stride 2 with three convolution layers. The three fully connected layers (FCL) are found in the final stage (FCLs). Furthermore, an activation function named SoftMax is used for the classification of skin cancer using dermoscopic images [28]. The Vgg 19 architecture is deeper than the Vgg16 which made it more complex and made training the model more costly [29]. The 50-layer residual network ResNet-50 [30] features have four-step architecture in which three deep residual networks are included with a kernel of 1, 64, 64 and 256, respectively. Furthermore, for the classification of diseases, a modified variant of Inception known as Inception v3 is commonly used by researchers. In the medical field, these pre-trained classifiers are widely utilized for disease classification. In addition, researchers consider these classifiers to be the most advanced classifiers currently available [44].

### 3.4. Proposed Architecture

The proposed SCDNet is based on the pre-trained Vgg16 model, which is then used by CNN’s network to classify multiple types of skin cancer using dermoscopy images. The architecture of CNN is based on three different layers which consist of the convolution layer, fully connected layer (FCL) and pooling layer (PL).

Vgg16 is followed by two blocks of CNN for feature extraction (FE). Figure 3 shows the model of SCDNet. The proposed model is trained by inputting the dermoscopy images of 224 × 224 resolution. The input images contain the RGB channels. The convolutional layer is the initial layer in our model. This layer initiates the process by employing filters, which are also known as the kernel. As seen in Equation (1), the size of the kernel is determined by two variables.
(1)$$Size\;of\;filter\left(SF\right)=F_{𝓌}\times F_{\ell }$$
where the filter width is denoted by $F_{𝓌}$ and the filter height is denoted by $F_{\ell }$. In the course of our research, we decided to keep the size of the filter at its default setting of 3, therefore Equation (1) is rewritten as SF = 3 × 3. These filters are also known as feature identifiers. These identifiers assist us in the extraction of low-level features of the images, which are also known as edges and curves [50]. The model has three additional convolution layers to extract deep features and produce complete patterns from dermoscopic images. Additionally, these filters initiate the convolution operations on the sub-area of dermoscopy images. The process of convolution multiplication and addition to filer is applied to the pixel values of dermoscopic images. The receptive field is another name for the dermoscopic images sub-area. The capability of the model to extract the feature components was increased gradually by adding more convolution layers to it.

When the model is trained, it learns the filter weights which are numerical values assigned to each filter element. Filtering starts the convolution process from the beginning of the dermoscopic image and continues through the whole image without interruption. The filter convolution technique concludes after the whole image has been processed [57]. A wide range of values is obtained from the feature maps of dermoscopic images [58]. The feature map values were calculated using Equation (2).
(2)$$F\left[g,\;h\right]=\left(SF\times I\right)\left[g,h\right]=\underset{j}{\sum }\underset{k}{\sum }SF\;\left(j,\;k\right)I\;\left[g+j,\;h+k\right]$$

The feature map is denoted by F, the input image is denoted by I, and the kernel is denoted by SF. The dimensions of the filters were determined by using j and k. The resulting array indexes were g and h. The value of the filter’s movement was limited by an additional parameter known as the stride. All convolution layers in our research have a stride value set to 1. Moreover, higher stride values lower the spatial dimension and cause difficulties, such as the receptive field exceeding the input size. Therefore, to solve these issues, a zero-padding strategy was used. This technique places padding of zero around the edge of the image so that the dimension of the output remains the same as the dimensions of the input image [59]. The zero-padding approach was used to calculate the stride value, as shown in Equation (3).
(3)$$Z_{p}=\frac{w-1}{2}$$
where $\;Z_{p}$ stands for zero paddings and the width of the filter is represented by w. The filter’s height and width are the same in our research. For extracting numerous and dominating features, Conv layers have utilized various filters. Our model’s initial convolution layer has a total of sixteen filters. However, the filter count of the remaining Conv layers was enhanced from 16 to 512. Equations (4)–(6) were used to calculate the output value also known as the activation map.
(4)$$\;Q_{w}=\frac{\;Im_{𝓌}-{ℱ}_{𝓌}+2Zp}{S}+1\;$$
(5)$$\;Q_{h}=\frac{\;Im_{h}-{ℱ}_{h}+2Zp}{S}+1\;$$
(6)$$\;Q_{i}={ℱ}_{n}\;$$
where the input image’s height and width are represented by $\;Im_{h}$ and $\;Im_{w}$, while the filter’s width and height are represented by $\;{ℱ}_{w}$ and $\;{ℱ}_{h}$. Zp represents the zero-padding and S shows the stride, whereas ${ℱ}_{n}$ shows the number of filters applied to the convolution layers. The initial convolution layer of our proposed method contained the following parameters: $\;Im_{w}$ = 224, $\;Im_{h}$ = 224, $\;{ℱ}_{w}$ = 3, $\;{ℱ}_{h}$ = 3, S = 1, $\;Z_{p}$ = 0, and ${ℱ}_{n}$ = 16. After putting the values into Equations (4)–(6) following results are obtained:$$\;Q_{w}=\frac{224-3+2\left(0\right)}{1}+1=222$$
$$\;Q_{h}=\frac{224-3+2\left(0\right)}{1}+1=222$$
$$\;Q_{i}=16$$

To start the process, the Conv outputs were corrected using a rectified linear unit (ReLU). ReLU activation replaces negative outcomes with zero. ReLU is used in our proposed CNN model to boost nonlinearity and computational time without compromising the accuracy of the model [45]. Adding the max-pooling layer (MP) after the convolution layers helps to lessen the spatial dimension of the input image. The filter size in SCDNet was set to 2 × 2, and for all MP layers, the value of stride was 2.

This filter applies a convolving effect to the whole input volume, which results in the greatest value possible for the sub-area of the image. It is noted that MP layers contain the location of one feature relative to another. It also minimizes computational cost by reducing weights and avoids the overfitting of the model. Then, the dropout layer was introduced. We tested our model with dropout values of 0.05, 0.20, and 0.25. However, only 0.20 proved significant. Different dropout values were used to avoid the overfitting of our proposed model [30]. This layer eliminated random activation and confirmed the model’s ability to predict the label. The one-dimensional feature vector was created by applying the flattened layer of the proposed model, which was used to transform the two-dimensional feature map. The flattened layer’s output was supplied to the FCL. The FCL used the one-dimensional feature vector to accomplish the classification process. The FCL used in this research has 512 neurons. The first dense layer of FCL transmits the activation outcome to the second dense layer. The final output of the proposed model is produced by a dense layer that consists of a soft-max activation function and four neurons. This layer is responsible for classifying the output image as belonging to one of the skin diseases: Melanoma, Melanocytic Nevi, Basal Cell Carcinoma, and Benign Keratosis.

Total parameters are 13,800,600, out of which 13,800,500 are trainable parameters (TP) and there are 100 non-trainable parameters. When it comes to determining the ideal value of a parameter, TPs are those that vary throughout training and must be used, while NTPs are those that remain unchanged throughout the process. Consequently, NTPs will not contribute to the categorization step. The description of the layer utilized in our proposed SCDNet is shown in Table 2.

### 3.5. Performance Evaluation

In this research the issues of multiclassification are resolved. Melanoma, Melanocytic Nevi, Basal Cell Carcinoma, and Benign Keratosis were classified correctly. The effectiveness of the model was evaluated using a confusion matrix. The confusion matrix is shown in Figure 4. Table 3 lists the confusion matrix’s parameters, as seen in Figure 3. The efficiency of our proposed SCDNet was evaluated using four metrics: accuracy (Accu), precision (P), sensitivity (SEN) also known as recall(R) and F1-score (FS). The following formulae, Equations (7)–(10), were used to calculate the values of these metrics.
(7)$$Accu=\frac{\;T_{P}+T_{N}}{\;T_{P}+F_{P}+T_{N}+F_{N}}\;$$
(8)$$\;\;Sen=\frac{\;T_{P}}{\;T_{P}+F_{N}}\;$$
(9)$$Pre=\frac{\;T_{P}}{\;T_{P}+F_{p}}\;$$
(10)$$\;\;\;FS=\frac{P\times R}{P+R}\times 2\;$$
where $\;T_{P}$ represents the True Positive, $\;T_{n}$ represents the true negative while $\;F_{N}$ and $F_{p}$ represent the False Positive and False Negative. The variables of Equations (7)–(10) are described in Table 4.

## 4. Results

The dermoscopic images are used as input for the Vgg16 + CNN-based SCDNet which was developed to classify various chest diseases. Using grid search techniques, several hyperparameters of the proposed method were fine-tuned to achieve optimal performance. The learning rate, the number of epochs, and the batch size are the included hyperparameters. Training of the SCDNet was completed in 50 epochs. A “stochastic gradient descent” (SGD) optimizer was used to set the starting learning rate for the four transfer learning models and SCDNet to 0.05. The momentum was adjusted to 0.8. The learning rate decreased by 0.1 after 10 epochs. This was done to avoid overfitting. Our SCDNet batch size was 20. The accuracy, precision, recall, ROC curve, confusion matrix, and F1-score were determined for each class label for the proposed SCDNet, ResNet-50, inceptionv3, AlexNet and Vgg19 architecture.

### 4.1. Experimentation Process

The Keras library was used to implement the SCDNet and the four pre-trained models. Python language was used to program the methods that were not connected to the Conv network. The experimentation was conducted on a Core i7, 8th gen windows-based machine with a 16 GB EVGA GeForce RTX GPU and 32 GB RAM.

### 4.2. SCDNet Results

The accuracy of SCDNet’s training and validation throughout the 50 epochs is shown in Figure 4. It has been observed that the maximum training accuracy acquired was 98.78%, whereas the highest possible level of validation accuracy obtained was 92.15%. The model’s training and validation losses were 0.011 and 0.069. These results demonstrated that our proposed SCDNet was properly trained and is capable of accurately identifying many skin cancers, including Melanoma, Melanocytic Nevi, Basal Cell Carcinoma, and Benign Keratosis.

The proposed SCDNet and four pre-trained models were evaluated using several performance metrics for the classification of Melanoma, Melanocytic Nevi, Basal Cell Carcinoma, and Benign Keratosis. To train, validate, and test the model, the dataset was segmented as follows: 70:20:10. A total of 25,331 dermoscopic images of various types of skin cancer are included in the dataset. The dataset includes 4522 images of Melanoma, 12,875 images of Melanocytic Nevi, 3323 images of Basal Cell Carcinoma, and 2624 images of Benign Keratosis. The classification accuracy of transfer learning classifiers and the proposed SCDNet was evaluated using these images. There are rows and columns in the confusion matrix (Figure 4) that reflect actual cases and predicted cases respectively. The confusion matrix of the proposed SCDNet and the transfer learning classifiers are shown in Figure 5.

The proposed method correctly classifies 1970 images of melanotic nevi out of 2099 total images while it misclassifies 38 images as basal cell carcinoma, 75 as melanoma and 16 images as benign keratosis. While the proposed model correctly classifies 550 images of basal cell carcinoma out of 622 total images, it misclassifies the 25 images as melanoma, 35 images as melanotic nevi and 12 images as benign keratosis. The proposed method correctly classifies 1620 images of melanoma out of 1782 images while it misclassifies the 90 images as melanotic nevi, 54 images as basal cell carcinoma and 18 images as benign keratosis. The SCDNet correctly classify 530 images of benign keratosis out of 563 images. It misclassifies 18 images as melanotic nevi, 3 images as basal cell carcinoma and 15 images as melanoma. However, the Resnet 50 correctly classifies 1955 images of melanotic nevi, 541 images of basal cell carcinoma, and 1600 images of melanoma and correctly classifies 525 images of benign keratosis. In comparison to this, Alex net correctly classifies 1798 images of melanotic nevi, 480 images of basal cell carcinoma, 1511 images of melanoma and 515 images of benign keratosis. The Vgg19 correctly classifies 1901 images of melanotic nevi, 487 images of basal cell carcinoma, 1480 images of melanoma and 492 images of benign keratosis. Furthermore, Inception-v3 correctly classifies 1835 images of melanotic nevi, 471 images of basal cell carcinoma, 1470 images of melanoma and 401 images of benign keratosis. 

As shown in Table 5, our proposed model achieved excellent results for the classification of skin cancer classes by achieving an accuracy of 96.91%, recall of 92.18%, precision of 92.19%, and achieving f1 score of 92.18%. The accuracy, recall, precision, and f1-score achieved by the Resnet-50 were 95.50%, 91.16%, 91.18%, and 91.00%, respectively. However, the Vgg19 model attained an accuracy of 94.25%, precision of 89.71%, recall of 89.20%, and f1-score of 89.44%. The Alexnet attained an accuracy of 93.10%, precision of 88.41%, recall of 88.32%, and f1-score of 88.36%. It has also been observed that Inception-v3 delivered worse outcomes compared to its competitor techniques. In conclusion, the classification accuracy of our proposed SCDNet is superior to that of the four existing transfer learning classifiers. Moreover, the performance of the proposed method is validated by applying for the leave one out cross validation (LOOCV). A single iteration of the learning process was performed on each instance, with the remaining instances serving as the training set and a single instance is chosen to serve as the test set. The 99% of the data was set for training purposes whereas 1% of the data was used for testing purposes. The results of LOOCV are given in Table 5.

All of these pre-trained classifiers are made up of deep neural networks, and the spatial resolution of the feature map that they use is taken from their most recent convolution layer. As a consequence, the resolution of the feature map has been substantially decreased, which has resulted in a reduction in the accuracy of their classification. In addition, the size of the filter for specific disease classification is also inappropriate; as a result, these pre-trained classifiers overlooked essential aspects and generated a large number of input receptive fields of neurons. The proposed SCDNet addresses the issues of poor resolution and overlapping prominent features present in the specific area of skin cancer-infected dermoscopic images. Our approach additionally incorporates the large size of the filters and speeds up the convergence while greatly decreasing the effect of noise which results in the enhancement of the classification performance. The model is deemed adequate and effective if it attained the largest area under the curve (AUC) of receiver operating characteristic (ROC). True-positive rate (TPR) and false-positive rate (FPR) are used to determine the ROC curve. The proposed method achieves an AUC(ROC) of 0.9516 whereas, the AUC was 0.9335 for Resnet, 0.9423 for Vgg19, 0.9226 for Alexnet, and 0.9183 for Inception v3. The AUC (ROC) findings reveal that our proposed SCDNet classifier beats the four existing classifiers. In addition, our SCDNet method helps healthcare professionals in the identification of skin cancer patients using dermoscopy images.

### 4.3. Comparison with the Most Advanced Classifiers

The comprehensive analysis of the proposed SCDNet in terms of accuracy, recall, f1 score and precision is performed with the most advanced classifiers available mentioned in Table 6.

Mijwil et al. [35] provide a Convnet net model based on inception v3 which works on the binary classification of skin disease. This method classifies skin cancer as benign or malignant. Dorj et al. [37] work on the multiclass classification of skin cancer and it achieves an accuracy of 92.83% for the classification of four classes of skin cancer which include Actinic Keratoses, Basal Cell Carcinoma, Squamous Cell Carcinoma and Melanoma. Afza et al. [38] utilize 2D superpixels with Resnet 50 for the multiclassification of skin lesions and achieves an accuracy of 85.50%. Khan et al. [42] also achieve an accuracy of 88.50% for the multiclassification of skin cancer. Chaturvedi et al. [51] achieve an accuracy of 92.83% for the multiclassification of skin cancer. In comparison to this, Zhang et al. [60] and Lie et al. [61] achieve an accuracy of 86.80% and 87% for the binary classification of skin cancer. The proposed method achieves an outstanding accuracy of 96.91% as compared to other state-of-the-art methods.

## 5. Discussion

Dermoscopy images are utilized for the screening and categorization of several skin cancers. Our method offers a complete view of a particular area, allowing us to identify the disease and interior affected areas. Dermoscopy is a more dependable and effective approach for detecting Melanoma, Melanocytic Nevi, Basal Cell Carcinoma and Benign Keratosis. A computerized diagnostic technique is required to identify the deadly melanoma since the number of confirmed cases is steadily rising. Dermoscopy images can automatically discriminate between melanoma positive patients and other skin cancer disorders using deep learning (DL) methods. As a result, we developed a deep learning-based SCDNet model that accurately identifies multiple skin diseases such as Melanoma, Melanocytic Nevi, Basal Cell Carcinoma and Benign Keratosis and enables healthcare practitioners to initiate the treatment for these patients at an earlier stage. The aforementioned experimental study demonstrates that our suggested SCDNet is effectively and substantially trained on skin cancer categories of Melanoma, Melanocytic Nevi, Basal Cell Carcinoma and Benign Keratosis and correctly classifies these infected cases. Our SCDNet model outperforms the other four pre-trained classifiers for the multi-classification of skin cancer. The proposed system achieves outstanding performance as compared to four pre-trained classifiers. Moreover, it achieves an accuracy of 92.18% for the classification of Melanoma, Melanocytic Nevi, Basal Cell Carcinoma and Benign Keratosis using dermoscopy images. The resolution of images is set to 224 × 224 × 3 for our proposed SCDNet and four transfer learning classifiers which include Inception v3, Vgg 19, AlexNet and Resnet 50. The proposed method was trained via a cross-entropy loss function. The classification performance of the proposed SCDNet is compared with the four transfer learning classifiers in Table 5. It has been noted that the model that we have presented gives an outstanding performance. It achieves an AUC of 0.9833, recalls of 99.9%, precision of 92.21%, an f1-score of 91.37%, and accuracy of 92.21%. The diagnostic performance of other competitors’ transfer learning methods with pre-trained weights is below that of SCDNet. Furthermore, Inception-v3 showed an AUC of 0.8245, recall of 83.42%, precision of 84.44%, f1-score of 84.32%, and accuracy of 83.23% which are the lowest when compared with the other models. The binary classification dilemma was unaffected by the use of pre-trained architectures such as CNNs. These pre-trained classifiers showed better performance for segmentation or identifying a disease from multiple classes [51,62]. Several studies [63,64,65] claim that the performance of pre-trained network decreases for binary classification tasks when the number of CNN layers increases. The proposed SCDNet efficiently identifies the pattern of anomalies and generates the discriminative sequences which help in the diagnosis of multiple types of skin cancer and achieve an accuracy of 92.21%. Table 5 presents the results obtained from the evaluation of the various pre-trained classifiers. In this study multi-classification of skin cancer is performed in which Melanoma, Melanocytic Nevi, Basal Cell Carcinoma, and Benign Keratosis are accurately classified. 

Furthermore, we provide a comprehensive explanation for why the diagnostic performance of state-of-the-art methods is lower than our method. The architecture of the pre-trained classifiers is based on deep networks, in which the spatial resolution of the final convolution layer contains fewer feature maps which limit the classification accuracy of the model. Other problems include networks with inappropriate filter sizes and an excessive number of input neurons that miss important features. These problems are solved by using our SCDNet model. We utilized a Vgg16-based CNN model with merged dilated convolution values for the classification of different types of skin cancers. Additionally, our SCDNet model has solved the issue of poor resolution and overlap in the infected area of the dermoscopy image. Additionally, our methodology speeds up the convergence while lowering the impact of structured noise which results in enhanced diagnostic performance. In the final step, we use the appropriate filter size of 3 × 3 for our proposed model. It is clear from the evaluation of experimental results that our proposed model for multi-classification of skin cancer using dermoscopy images is an effective tool for doctors.

## 6. Conclusions

There has been an increase in the number of people affected by melanoma and other forms of skin cancer throughout the globe in recent years. A speedy and effective diagnostic process is required because of the large number of cases. There have been a large number of deaths occurred due to late diagnosis. The number of deaths can be minimized if skin cancer is detected at an early stage. After the emergence of deep learning-based diagnostic systems, the traditional methods of diagnosis are becoming outdated. In this work, a multi-classification model named SCDNet was developed and analyzed for the diagnosis of multiple skin cancer diseases from dermoscopy images, including Melanoma, Melanocytic Nevi, Basal Cell Carcinoma, and Benign Keratosis. The CNN-based SCDNet can automatically identify prominent features in dermoscopy images. An exhaustive experiment demonstrates that SCDNet has the highest diagnostic performance when compared with the well-known pre-trained classifiers. From the results, we believe that the SCDNet has the potential to play an important part as a guiding hand for medical professionals. Moreover, the use of deep learning in dermoscopy systems also improves the quality and provides convenience for the user and a cost-effective solution for the diagnosis of skin cancer. The sole drawback of our study is that it cannot use an image dataset of dark skinned people for the diagnosis of skin cancer. In the future, we will use a pre-trained model which extracts features from other publicly available datasets such as ISIC 2020. 

## Figures and Tables

**Figure 1 sensors-22-05652-f001:**
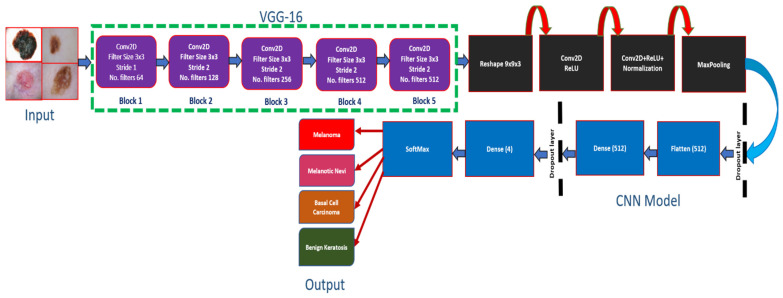
The architecture of Proposed SCDNet.

**Figure 2 sensors-22-05652-f002:**
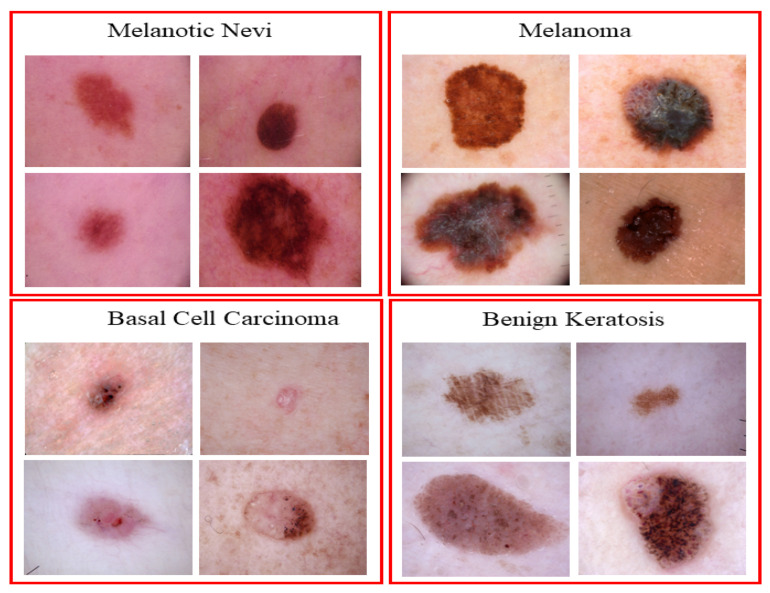
Sample images from the ISIC 2019 dataset.

**Figure 3 sensors-22-05652-f003:**
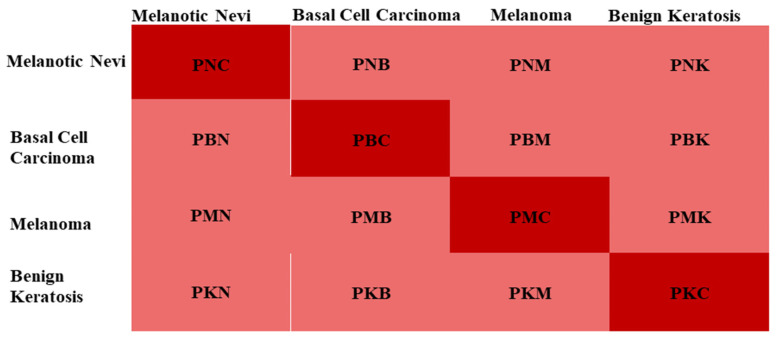
Confusion matrix of Proposed SCDNet.

**Figure 4 sensors-22-05652-f004:**
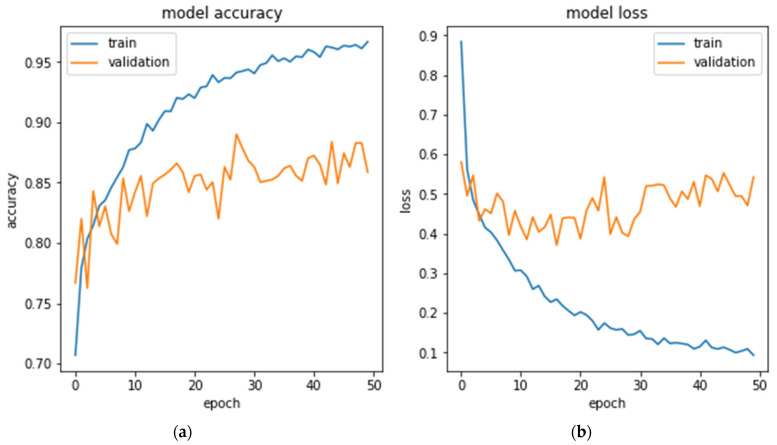
(**a**) SCDNet Model accuracy for training and validation (**b**) SCDNet Model loss for training and validation.

**Figure 5 sensors-22-05652-f005:**
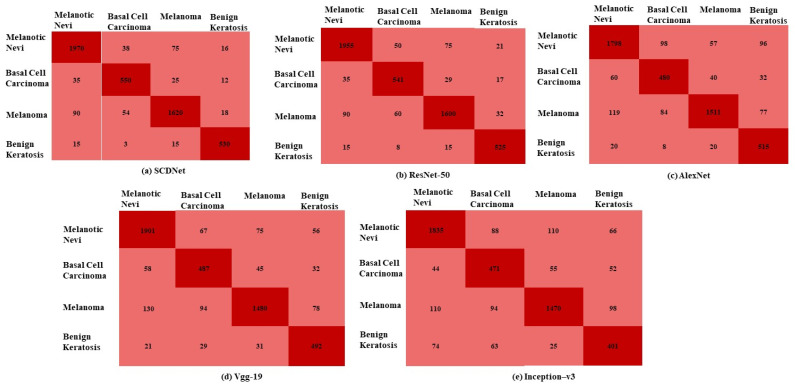
Confusion matrix for (**a**) SCDNet (**b**) Resnet 50 (**c**) Alexnet (**d**) Vgg-19 (**e**) Inception-v3.

**Table 1 sensors-22-05652-t001:** Dataset division into training, validation and testing sets.

Dataset Division	Melanoma (mel)	Melanocytic Nevi (nv)	Benign Keratosis (bk)	Basal Cell Carcinoma (bcc)	Total Images
Training Set	3166	9013	1837	2327	17,731
Validation Set	452	1287	263	332	2533
Testing Set	904	2575	524	664	5066
Total	4522	12,875	2624	3323	25,331

**Table 2 sensors-22-05652-t002:** Summary of proposed SCDNet.

Types of Layers	Shape	Parameters
Vgg16 (layers)	(7,7,512)	2,359,808
global average pooling (Reshape)	(5,5,512)	0
dropout (Droupout)	(3,3,512)	0
dense (Dense)	512	262,656
Dense (Dense)	4	2050
		264,708
Total params		264,708
Train params		2,359,808

**Table 3 sensors-22-05652-t003:** Confusion Matrix’s Parameters.

Parameters	Explanation
PMC	Melanoma correctly classified as Melanoma
PMN	Melanoma incorrectly classified as Melanocytic nevi
PMB	Melanoma incorrectly classified as Basal Cell Carcinoma
PMK	Melanoma incorrectly classified as Benign Keratosis
PNC	Melanocytic nevi is correctly classified as Melanocytic nevi
PNM	Melanocytic nevi incorrectly classified as Melanoma
PNB	Melanocytic nevi is incorrectly classified as Basal Cell Carcinoma
PNK	Melanocytic nevi is incorrectly classified as Benign Keratosis
PBC	Basal Cell Carcinoma is correctly classified as Basal Cell Carcinoma
PBM	Basal Cell Carcinoma is incorrectly classified as Melanoma
PBN	Basal Cell Carcinoma is incorrectly classified as Melanocytic nevi
PBK	Basal Cell Carcinoma is incorrectly classified as Benign Keratosis
PKC	Benign Keratosis is correctly classified as Benign Keratosis
PKM	Benign Keratosis incorrectly classified as Melanoma
PKN	Benign Keratosis incorrectly classified as Melanocytic nevi
PKB	Benign Keratosis incorrectly classified as Basal Cell Carcinoma

**Table 4 sensors-22-05652-t004:** Equations for confusion matrix.

Labels	TP	TN	FP	FN
Melanoma	PMC	PNM + PNB + PMB + PBM + PBN + PMN + PMC + PBC + PNC	PKM + PKN + PKB	PBK + PMK + PNK
Melanocytic Nevi	PNC	PKB + PKM + PBK + PMK + PKC + PMB + PBM + PMC + PBC	PNK + PNM + PNK	PBN + PMN + PKN
Basal Cell Carcinoma	PBC	PBC + PNB + PKN + PKN + PNC + PBN + PBK + PNK + PKC	PMK + PMN + PMB	PKM + PNM + PBM
Benign Keratosis	PKC	PKC + PNK + PMK + PKN + PNC + PMN + PKM + PNM + PMC	PBM + PBN + PKB	PMB + PNK + PKB

**Table 5 sensors-22-05652-t005:** Performance comparison of SCDNet with pre-trained classifiers.

Classifier	Accuracy	Recall	Precision	F1-Score
SCDNet	96.91%	92.18%	92.19%	92.18%
SCDNET(LOOCV)	94.98%	91.35%	91.24%	91.30%
Resnet 50	95.50%	91.16%	91.18%	91.00%
Vgg-19	94.25%	89.71%	89.20%	89.44%
Alexnet	93.10%	88.41%	88.32%	88.36%
Inception-v3	92.54%	87.34%	87.36%	87.33%

**Table 6 sensors-22-05652-t006:** Performance comparison of SCDNet with pre-trained classifiers.

Model	Accuracy	Recall	Precision	F1-Score	Reference
ConvNet	86.90%	86.14%	87.47%	-----	[35]
ECOC SVM	93.35%	97.01%	90.82%	-----	[37]
2D superpixels + MASK-RCNN	85.50	83.40%	84.50%	85.30%	[38]
InceptionResnetV2 + ResNeXt101	88.50%	87.40%	88.10%	88.30%	[42]
Inception-v3	92.83%	84.00%	83.00%	84.00%	[51]
ARL-CNN	86.80%	87.80%	86.70%	-----	[60]
Densnet & Resnet	87.00%	-----	-----	-----	[61]
SCDNet	96.91%	92.18%	92.19%	92.18%	

## Data Availability

Not applicable.

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
