# Peer review of "SCDNet: A Deep Learning-Based Framework for the Multiclassification of Skin Cancer Using Dermoscopy Images"

_sensors, 2022, doi:10.3390/s22155652_

Round 1

Reviewer 1 Report

Dear Author

I think that the results are good in this manuscript.
Can you try to analysis combinations of some models? 

Author Response

< Reviewer 1 >

Dear, We appreciate your reviewing our paper. We thoroughly revised our paper according to the reviewer's comments, including English revision through premium English editing tools. All the revisions are shown in the revised paper (in red color).

< Comment 1 >

I think that the results are good  in this manuscript. Can you try to anlysis combination of some models.

< Author Response >

Thanks for the suggestion. We have proposed an independent model that is not combined with any other model, we have applied vgg 16 for feature extraction and classification of these features is performed by using the four last layers of CNN. In the future, we will perform ensembling by combining different models for feature extraction and classification.

Reviewer 2 Report

Line 102: Benign Keratosis skin cancers: this is incorrect and incompatible with the expertise and knowledge of the colleagues that authored the paper. 

Please edit the manuscript for use of the English language, text, syntax, and intended meaning. 

Line 109: Given the claims are significant, would it be possible to train and evaluate  the proposed model in more than one ways?

Line 121: A considerable amount of research has been conducted for the diagnosis of skin cancer saving the time and the effort of the healthcare experts.: this first sentence is very awkward and redundant. Similarly, a number of other sentences would benefit from redrafting in my view. 

L217: "The This section": have the authors re-read the manuscript prior to submission?

Author Response

< Reviewer 2 >

Dear,

We appreciate your reviewing our paper. We thoroughly revised our paper according to the reviewer's comments, including English revision through premium English editing tools. All the revisions are shown in the revised paper (in red color).

< Comment 1 >

Line 102: Benign Keratosis skin cancer: this is incorrect and incompatible with the expertise and knowledge of the colleagues that authored the paper.  need to be in a different place, also the quality is not essential.

< Author Response >

Thank you for the suggestion. Sentence has been corrected with enhanced quality.

< Comment 2 >

Please edit the manuscript for the use of the English language, text, syntax and intended meaning.

< Author Response >

Whole manuscript has been proofread to avoid text, syntax and English language problems. Furthermore, we thoroughly revised our paper for English revision through premium English editing tools such as Grammarly.

< Comment 3 >

Line 109: Given the claims are significant, would it be possible to train and evaluate the proposed model in more than one ways?

< Author Response >

Thank for the suggestion. To train the proposed method in more than one way, leave one out cross validation (LOOCV) has been performed to evaluate the performance of proposed method.

< Comment 4 >

Line 121: A considerable amount of research has been conducted for the diagnosis of the skin cancer saving the time and effort of the healthcare experts: this foirst sentence is very awkward and redundant. Similarly a number of other sentences woiuld benefit from redrafting in my view.

< Author Response >

Thank you for the suggestion. The sentence has been changed. Moreover, changes have been made in the whole manuscript.

< Comment 5 >

Line 217: “The This section”: have the authors re-read the manuscript prior to submission?

< Author Response >

Repeating words has been removed from the whole manuscript.

Reviewer 3 Report

Summary: Skin cancer is the deadliest disease, and its early diagnosis enhances the chances of survival. Deep learning algorithms for skin cancer detection have become popular in recent years. A novel framework based on deep learning is proposed in this study for the multiclassification of skin  cancer types such as Melanoma, Melanocytic Nevi, Basal Cell Carcinoma and Benign Keratosis. The proposed model is named as SCDNet which combines Vgg-16 with convolutional neural networks (CNN) for the classification of different types of skin cancer. Moreover, the accuracy of the proposed method is also compared with the four state-of-the-art pre-trained classifiers in the medical domain named Resnet 50, inception v3, AlexNet and Vgg19. The performance of the proposed SCDNet classifier, as well as the four state-of-the-art classifiers, is evaluated using the ISIC 2019 dataset. The accuracy rate of the proposed SDCNet is 96.91% for the multiclassification of skin cancer whereas, the accuracy rates for Resnet 50, Alexnet and Vgg-19 and Inception-v3 are 95.21%, 93.14%, 94.25% and 92.54% respectively. The results showed that the proposed SCDNet performed better than the competitor classifiers.

I suggest the following minor revision: 

1) "The Skin cancer is caused by the uncontrolled growth of abnormal skin cells which results in malignant tumors. "- add reference for this statement. 

2) "Dermoscopy is a common technique to detect skin cancer. However, the initial appearance of multiple types of skin cancers is the same so it is very challenging for the dermatologist to identify them accurately"- add reference for this statement. 

3) Add the facts and figures of skin cancer for year 2022 and 2021. 

4) Explain the clinical methods with reference article such as ABCD rule, ABCDE rule and 7 point check list. 

5) "Melanoma is the most common and fatal type of skin cancer which is caused by irregular melanin growth in the cells of melanocytes. "- add reference for this statement.

6) "Melanin is typically present in the epidermal layer in benign lesions (common nevi). In the malignant lesion, melanin is produced at an abnormally high rate. Every year more than 5 million new cases of skin cancer are registered in the United States. . "- add reference for this statement. 

7) Add the dark sides of this work under the conclusion section. Also explain the hyper parameters choice of this work. 

Author Response

< Reviewer 3 >

Dear,

We appreciate your reviewing our paper. We thoroughly revised our paper according to the reviewer's comments, including English revision through premium English editing tools. All the revisions are shown in the revised paper (in red color).

< Comment 1 >

1) “The Skin cancer is caused by the uncontrolled growth of abnormal skin cells which results in malignant tumors”- add reference for this statement.

< Author Response >

Thanks for the suggestion, Reference has been added.

< Comment 2 >

2) “Dermoscopy is the common technique to detect skin cancer. However, initial appearance of multiple types of skin cancer is same so it is very challenging for the dermatologist to identify them accurately” - add reference for this statement.

< Author Response >

Thanks for the suggestion, Reference has been added.

< Comment 3 >

3) Add facts and figures of skin cancer for year 2022 and 2021.

< Author Response >

Thanks for the suggestion. Facts and figures for the years 2021 and 2022 are added in the Introduction.

< Comment 4 >

4) Explain the clinical methods with reference articles such as ABCD rule, ABCD rule and 7-point check list.

< Author Response >

ABCD rule and ABCD rule with 7-point checklist are added in the article with references [21] and [22] added in the Introduction.

< Comment 5 >

5) “Melanoma is the most common and fatal type of skin cancer which is caused by the irregular melanin growth in the cells of melanocytes” - add the reference for this statement

< Author Response >

Thanks for the suggestion, Reference has been added.

< Comment 6 >

6) “Melanin is typically present in the epidermal layer in benign lesions (common nevi). In the malignant lesion, melanin is produced at an abnormally high rate. Every year more than 5 million new cases of skin cancer are registered in the united states” - add the reference for this statement.

< Author Response >

Thanks for the suggestion, Reference has been added.

< Comment 7 >

7) Add the dark sides of this work under the conclusion section. Also, explain the hyperparameter choice of this work.

< Author Response >

The limitation of this study is added in the conclusion. The hyperparameter choice for this work is mentioned in sections 3.4 and 4.2.

Round 2

Reviewer 2 Report

Thank you for addressing the queries and comments.